# A Mutated Prostatic Acid Phosphatase (PAP) Peptide-Based Vaccine Induces PAP-Specific CD8^+^ T Cells with Ex Vivo Cytotoxic Capacities in HHDII/DR1 Transgenic Mice

**DOI:** 10.3390/cancers14081970

**Published:** 2022-04-13

**Authors:** Pauline Le Vu, Jayakumar Vadakekolathu, Sarra Idri, Holly Nicholls, Manon Cavaignac, Stephen Reeder, Masood A. Khan, Dennis Christensen, Alan Graham Pockley, Stéphanie E. McArdle

**Affiliations:** 1The John van Geest Cancer Research Centre, School of Science and Technology, Nottingham Trent University, Nottingham NG11 8NS, UK; pauline.levu@abzena.com (P.L.V.); jayakumar.vadakekolathu@ntu.ac.uk (J.V.); sarra.idri2014@my.ntu.ac.uk (S.I.); holly.nicholls2016@my.ntu.ac.uk (H.N.); manon.cavaignac@inserm.fr (M.C.); stephen.reeder@ntu.ac.uk (S.R.); graham.pockley@ntu.ac.uk (A.G.P.); 2Centre for Health, Ageing and Understanding Disease (CHAUD), School of Science and Technology, Nottingham Trent University, Nottingham NG11 8NS, UK; 3Department of Urology, University Hospitals of Leicester NHS Trust, Leicester LE1 5WW, UK; masood.khan@uhl-tr.nhs.uk; 4Center for Vaccine Research, Statens Serum Institute, DK-2300 Copenhagen, Denmark; den@ssi.dk

**Keywords:** prostate cancer, castrate (hormone)-resistant PCa (CRPC), prostatic acid phosphatase (PAP), CAF^®^09, immunotherapy, PBMC, HHDII/DR1 mice, Dextramer™

## Abstract

**Simple Summary:**

Treatments for castrate (hormone)-resistant prostate cancer (CRPC) remain limited and are not curative. Although the first (and only) FDA-approved vaccine for CRPC (PROVENGE^®^, Sipuleucel-T) has been shown to improve the overall survival of patients, it is not curative and its cost prevents its widespread use. PROVENGE^®^ induces immunity to prostatic acid phosphatase (PAP), a protein which is highly expressed in prostate cancer (PCa). Herein, we have developed a new PAP-based vaccine for PCa and demonstrated the presence of circulating PAP-specific CD8^+^ T cells that are responsive to this vaccine in patients with PCa. We have also shown that this new PAP sequence-derived peptide containing a modified amino sequence, in association with a strong adjuvant called CAF^®^09, induces strong immune responses and cytotoxic potential in a murine HHDII/DR1 transgenic model.

**Abstract:**

Background: Current treatments for castrate (hormone)-resistant prostate cancer (CRPC) remain limited and are not curative, with a median survival from diagnosis of 23 months. The PAP-specific Sipuleucel-T vaccine, which was approved by the FDA in 2010, increases the Overall Survival (OS) by 4 months, but is extremely expensive. We have previously shown that a 15 amino accid (AA) PAP sequence-derived peptide could induce strong immune responses and delay the growth of murine TRAMP-C1 prostate tumors. We have now substituted one amino acid and elongated the sequence to include epitopes predicted to bind to several additional HLA haplotypes. Herein, we present the immunological properties of this 42mer-mutated PAP-derived sequence (MutPAP42mer). Methods: The presence of PAP-135-143 epitope-specific CD8^+^ T cells in the blood of patients with prostate cancer (PCa) was assessed by flow cytometry using Dextramer™ technology. HHDII/DR1 transgenic mice were immunized with mutated and non-mutated PAP-derived 42mer peptides in the presence of CAF^®^09 or CpG ODN1826 (TLR-9 agonist) adjuvants. Vaccine-induced immune responses were measured by assessing the proportion and functionality of splenic PAP-specific T cells in vitro. Results: PAP-135-143 epitope-specific CD8^+^ T cells were detected in the blood of patients with PCa and stimulation of PBMCs from patients with PCa with mutPAP42mer enhanced their capacity to kill human LNCaP PCa target cells expressing PAP. The MutPAP42mer peptide was significantly more immunogenic in HHDII/DR1 mice than the wild type sequence, and immunogenicity was further enhanced when combined with the CAF^®^09 adjuvant. The vaccine induced secretory (IFNγ and TNFα) and cytotoxic CD8^+^ T cells and effector memory splenic T cells. Conclusions: The periphery of patients with PCa exhibits immune responsiveness to the MutPAP42mer peptide and immunization of mice induces/expands T cell-driven, wild-type PAP immunity, and therefore, has the potential to drive protective anti-tumor immunity in patients with PCa.

## 1. Introduction

Prostate cancer (PCa) is the second most frequent cancer in men and the fifth most frequent cause of cancer-related deaths in men worldwide [1]. Patients with organ-confined PCa are typically treated with local therapies, such as external beam radiotherapy or radical prostatectomy, or managed by active surveillance [2]. However, one third of these patients develop disease recurrence or metastasis and require androgen deprivation therapy (ADT) [3]. ADT has a time-limited efficacy, as patients eventually become resistant to treatment and develop castrate (hormone)-resistant PCa (CRPC). Patients with CRPC exhibit a median overall survival (OS) from diagnosis of 23.2 months in the absence of metastasis, or 13.3 months in the presence of metastasis [4,5]. Treatments for CRPC include androgen signaling inhibitors (Enzatulamide, Abiraterone), chemotherapy (Carbazitaxel, Docetaxel), the radiopharmaceutical agent radium-223, or the Sipuleucel-T vaccine [6].

Sipuleucel-T, which was approved by the FDA in 2010, is recommended for early-stage CRPC (Gleason score 7) and has been shown to prolong the Overall Survival (OS) by a median of 4.1 months, but to have no effect on progression-free survival (PFS) [7]. Sipuleucel-T comprises ex vivo stimulation of the patient’s peripheral blood mononuclear cells (PBMCs) with a fusion protein composed of the prostatic acid phosphatase (PAP) protein and granulocyte/monocyte colony stimulating factor (GM-CSF), followed by the re-infusion of these cells into the patient. Although this approach has been demonstrated to induce immunity towards PAP, it remains expensive and of limited efficacy. Moreover, it is well-established that peptides are better at inducing CD8^+^ T cell responses than whole proteins [8]. New therapies for the treatment of CRPC are, therefore, urgently needed. With this in mind, many of the clinical trials of new therapies for advanced PCa that are currently recruiting or recently completed use immunotherapeutic approaches (Table 1), with some targeting specific antigens such as prostate-specific membrane antigen (PSMA), prostate stem cell antigen (PSCA), PAP, NY-ESO-1, MAGE-C2, Mucin 1, and Bcl-xl (the latter in combination with the CAF^®^09b adjuvant). One approach uses a cocktail of HLA-A2 peptides derived from tumor-associated antigens in combination with DPX-0907, a polynucleotide-based adjuvant and a universal Th peptide. However, to date, Sipuleucel-T remains the only vaccine approach to improve the OS of patients with PCa.

The relative restriction of PAP expression to prostate tissue [9] and its expression in 95% of primary prostate tumors make it a good candidate on which to base PCa vaccines [10].

Previous work conducted in our laboratory demonstrated that vaccination of mice with a 15 mer PAP-derived peptide containing an HLA-A*02:01 class-I restricted epitope within an HLA-DRB1*01:01 class-II-restricted epitope induces PAP-specific T cell responses [11]. This 15 mer PAP-derived vaccine, when administered as a DNA vaccine, reduced tumor growth in a syngeneic heterotopic murine model of PCa. Given that long synthetic peptides enable multiple epitopes to be incorporated and broadens their applicability from HLA-A2^+^ patients to a larger proportion of the population [12] and that longer peptides induce stronger CD8^+^ T cell reactivity in vivo and a more efficient and robust protective immune response [13], we elongated the peptide to 42 amino acids. We also ‘mutated’ the natural sequence and found that a change of amino acid at position 116 from alanine to leucine increased the predicted MHC binding score of several epitopes.

The aim of the present study was to confirm the presence of circulating PAP-specific CD8^+^ T cells after stimulating PBMCs from patients with PCa using either the full MutPAP42mer or a MutPAP42mer-derived 9 mer epitope with Dextramer™-based flow cytometry. Thereafter, we assessed the potential of this novel MutPAP42mer as a vaccine for eliciting strong PAP-specific T cell immune responses. The study compared the immunogenicity of the MutPAP42mer peptide vaccine with that of its wild-type counterpart and the influence of CpG and CAF^®^09 adjuvants thereupon. CpG oligonucleotides (ODN) target Toll like receptor (TLR)-9 expressed on dendritic cells (DCs) and B lymphocytes, and induce the production of pro-inflammatory cytokines, leading to improved antigen presentation and generation of vaccine-specific cellular responses of the Th1 type [14]. CAF^®^09 is an adjuvant based on cationic liposomes incorporating the synthetic analog of C-type lectin receptor agonist Monomycoloyl Glycerol (MMG)-analog MMG-1 and the TLR-3 agonist poly(I:C). This adjuvant induces the production of pro-inflammatory cytokines and chemokines, as well as type-1 IFNs, thereby leading to strong CD4^+^ Th1 cell and CD8^+^ cytotoxic T cell responses [15].

## 2. Materials and Methods

### 2.1. Mouse Strain

HHDII/DR1 double transgenic mice expressing human α1 and α2 chains of HLA-A*0201 chimeric with α3 chain of H-2Dd allele (HHDII), also expressing HLA-DRB*0101, and knocked out for the expression of murine MHC class I (H-2b) and II (I-Ab) were provided by Dr. Lone (CNRS, Orleans, France). These were bred at Nottingham Trent University. The animal study protocol was approved by the Institutional Review Board (Animal Welfare Ethical Review Body, AWERB) at Nottingham Trent University. All animals were housed and handled in accordance with the UK Home Office Codes of Practice for the Housing and Care of Animals. The pre-clinical studies were approved by the UK Home Office under the Animals (Scientific Procedures) Act 1986 (PPL 40/3563 expired 5 December 2016; PB26CF602 valid 28 November 2016—28 November 2021).

Males were used and treatment started when they were between 7 and 18 weeks of age. In HHDII/DR1 mice, murine MHC class I and II has been replaced with the human HLA-A2 (chimeric monochain: HHD molecule: α1–α2 domains of HLA-A2.1, α3 to cytoplasmic domains of H-2 Db, linked at its N terminus to the C terminus of human β2m by a 15-amino acid peptide linker and human DR1 alleles) [16].

### 2.2. Cell Lines, Transfections, and Transductions

LNCaP human PCa cells (American Type Culture Collection, ATCC) are androgen-sensitive human prostate adenocarcinoma cells derived from a left supraclavicular lymph node metastasis. LNCaP cells were cultured in RPMI 1640 medium (Lonza via Scientific Laboratory Supplies (SLS) Limited, Nottingham, UK) supplemented with 10% *v*/*v* fetal bovine serum (FBS, ThermoFisher, Paisley, UK), 1% L-glutamine (SLS/Lonza, Nottingham UK), 1% HEPES (SLS/Lonza, Nottingham, UK), and 0.20% *w*/*v* glucose (Sigma-Aldrich, Merck Life Science UK Limited, Gillingham, UK). LNCaP cells were overexpressed for the expression of chimeric HHDII to match the alpha 3 loop and the transmembrane domain of the HHDII/DR1 mice. For this, cells were transfected at 50–60% confluency with 8 µg of plasmid of interest in Opti-MEM™ reduced serum medium, Lipofectamine™ P3000 and enhancer reagent. Successfully transfected cells were then selected in 1 mg/mL G418 (aminoglycoside antibiotic) (Sigma-Aldrich, Merck Life Science UK Limited, Gillingham, UK). The T2 lymphoma-derived cell line (ATCC) was cultured in RPMI 1640 medium supplemented with 10% *v*/*v* FBS and 1% L-glutamine. T2 cells express low amounts of HLA-A2 on the cell surface due to TAP deficiency and can only present/be loaded with exogenous peptides. Wild type (WT) and HHDII-transfected LNCaP cells were further transfected to knock down for the expression of the human PAP gene. For this, MISSION shRNA pLKO.1-puro plasmids in Bacterial Glycerol Stock were obtained from Sigma-Aldrich, Merck Life Science UK Limited, Gillingham, UK . Following glycerol stock bulk up in Laurie Broth medium containing 100 µg/mL ampicillin, plasmids were isolated using the QIAGEN Plasmid Midi Kit (Qiagen, Manchester, UK), according to the manufacturer’s protocol, and the concentration was measured on a Nanodrop ND™ Spectrophotometer. Plasmids were immediately stored at −20 °C. Lentiviral transduction was performed using HEK293t cells as packaging cells to produce viral particles. For this, HEK293t cells (ATCC) were grown in Dulbecco’s modified Eagle’s medium (DMEM) supplemented with 10% *v*/*v* FBS and 1% L-glutamine. HEK293t cells were transfected with Lipofectamine™ 3000, as described above, using three plasmids: MISSION shRNA pLKO.1-puro plasmid, packaging plasmid, and envelope plasmid. Viral particles were collected on day 3 after transfection and LNCaP cells were transduced at 70% confluency. Successfully transduced LNCaP cells were selected in 1 µg/mL puromycin (Sigma-Aldrich, Merck Life Science UK Limited, Gillingham, UK).

### 2.3. Healthy Donors, and Men with Benign Prostate Disease and PCa

Peripheral blood mononuclear cells (PBMCs) from healthy individuals were either purchased from CTL-Europe (Bonn, Germany) or isolated from whole blood collected from volunteers at Nottingham Trent University. Whole blood samples were obtained from men with benign prostate disease and PCa (Clinical Data, Table A3) attending the Urology Clinic at University Hospitals of Leicester NHS Trust, Leicester, UK by Professor Masood A. Khan (Consultant Urologist).

The study was conducted in accordance with the Declaration of Helsinki. Ethical approval for the collection of peripheral blood and the analysis of PBMCs from healthy donors and from men with benign prostate disease and PCa was provided by Nottingham Trent University’s Ethical Committee (Humans) and by the National Research Ethics Committee of the East Midlands (REC Ref: 18/WM/0377) and the Research and Development Department in the University Hospitals of Leicester NHS Trust, respectively. Informed consent was obtained from all donors.

PBMCs were isolated from whole blood by density gradient centrifugation using Lymphocyte Separation Medium (Sigma-Aldrich, Merck Life Science UK Limited, Gillingham, UK) and Leucosep™ tubes (Greiner Bio-One Ltd., Stonehouse, UK), HLA-A2 expression determined by flow cytometry using an APC-conjugated HLA-A2 monoclonal antibody (mAb, BioLegend, London, UK, clone BB7.2) and cryopreserved in liquid nitrogen until Dextramer™-based analysis and cytotoxicity assays. Briefly, 1 × 10^6^ PBMCs were incubated with the HLA-A2 mAb for 30 min at 4 °C in the dark, washed, and then resuspended in phosphate buffered saline (PBS). Samples were acquired on a 10-color Gallios™ flow cytometer (Beckman Coulter (UK) Limited, High Wycombe, UK) and analyses were performed using Kaluza™ software 1.3 (Beckman Coulter (UK) Limited, High Wycombe, UK).

### 2.4. Vaccine Reagents and Immunization Procedure

Peptides purchased from GenScript Biotech (Piscataway, NJ, USA) were resuspended at 10 mg/mL in 100% v/v dimethyl sulfoxide (DMSO) and stored at −80 °C. Sequences of the wild type (WT) and mutated human PAP42mer peptides were as follows:wtPAP42: YIRSTDVDRTLMSAMTNLAALFPPEGVSIWNPILLWQPIPVHmutPAP42: YIRSTDVDRTLMSLMTNLAALFPPEGVSIWNPILLWQPIPVH

The CpG ODN nucleotide (ODN 1826) was purchased from Eurofins and CAF^®^09 was produced by the Statens Serum Institut (Copenhagen, Denmark), as described previously (https://doi.org/10.3389/fimmu.2018.00898, accessed on 30 April 2018). CAF^®^09 is a liposomal cationic adjuvant formulation composed of dimethyldioctadecylammonium (DDA) bromide, the synthetic monomycoloyl glycerol (MMG) analog MMG-1, and the synthetic TLR3 agonist polyinosinic:polycytidylic acid [poly(I:C)] electrostatically adsorbed to the DDA headgroups.

The CpG-based vaccine was prepared by mixing 50 µg of CpG and 30 µg of PAP42mer peptide in a final volume of 100 µL sterile PBS (Lonza via Scientific Laboratory Supplies Limited, Nottingham, UK) and was administered by intramuscular injection at the base of the tail. The CAF^®^09-based vaccine was prepared by mixing 100 µL of CAF^®^09 and 30 µg of PAP42mer peptide in a final volume of 200 µL of PBS containing 9% *w*/*v* sucrose and this was administered by intraperitoneal injection. Animals were immunized as described three times with two-week intervals. One week after the final immunization, splenocytes were isolated and cultured in T cell medium (RPMI 1640 medium supplemented with 10% *v*/*v* FBS, 1% L-glutamine, 1% HEPES, 2% Pen/Strep antibiotic, and 0.005% fungizone).

### 2.5. Quantification of Peptide-Specific IFNγ-Secreting T Cells

Murine IFNγ ELISpot assays were performed according to the manufacturer’s protocol (Mabtec AB, Nacka Strand, Sweden) using 96-well ELISpot plates (Millipore, Merck Life Science UK Limited, Gillingham, UK). For each animal, splenocytes (0.5 × 10^6^ per well in T cell medium) were stimulated with 1 μg/mL of each 9 mer peptide or 10 μg/mL of each 15 mer peptide (Table 2). Control wells contained no peptides. Every condition was performed in triplicate wells. After 48 h, plates were developed for 15–30 min using BCIP NBT Substrate Solution (Bio-Rad Laboratories, Watford, UK), after which they were rinsed with tap water. Spots were quantified using an ELISpot reader (Cellular Technology Limited, Cleveland, OH, USA), with an average saturation of spot being counted of 800.

### 2.6. Flow Cytometry Analysis of Splenocyte Phenotype and Cytokine Secretion Profiles 

Splenocytes (0.5–1 × 10^6^) were stained by flow cytometry [17,18] as follows. Fc receptors were blocked by incubating with a CD16/CD32 mAb (BioLegend, UK, clone 93) for 15 min at 4 °C, after which they were incubated with panel of mAbs to surface antigens (Table A1) for 30 min at 4 °C in the dark. LIVE/DEAD™ Fixable Yellow (Life Technologies, ThermoFisher, Paisley, UK) was used to identify viable cells. Samples were acquired and analyzed as described in Section 2.3.

For intracellular cytokine staining, splenocytes (1 × 10^6^ cells/well in T cell medium) were stimulated with 1 µg/mL of ILLWQPIPV (referred to as ILL from there on) 9 mer peptide, 10 µg/mL of VSIW 15 mer peptide, or without peptide in the presence of co-stimulatory murine CD28 (BioLegend, London, UK, clone 37.51) and CD49d (BioLegend, London, UK, clone 9C10) mAbs (1 µg/mL) for 1 h at 37 °C in 96-well plates. Cells were then incubated with Brefeldin A (5 µg/mL, BioLegend, London, UK) and Monensin (2 µM, BioLegend, London, UK) and a FITC-conjugated CD107a mAb (BioLegend, London, UK, clone 1D4B)—a marker of degranulation—for a further 5 h at 37 °C. The plate was then kept at 4 °C overnight until staining the following day. After blocking and staining with the mAbs described above (Table A1), cells were fixed and permeabilized according to the PerFix-nc Kit protocol (Beckman Coulter (UK) Limited, High Wycombe, UK) and stained for intracellular expression of the cytokines TNF-α, IL-2, IFNγ and intranuclear expression of the cell proliferation marker Ki67). Samples were acquired and analyzed as described in Section 2.3.

### 2.7. Ex Vivo Expansion of Splenocytes

In order to increase the frequency of vaccine-specific CD8^+^ T cells, splenocytes from each immunized mouse were cultured in 24 well plates (5 × 10^6^/2 mL) for 6 days at 37 °C in T cell medium containing 50 U/mL of recombinant murine IL-2 (R&D Systems, Biotechne, Abingdon, UK), 2 mM of β-mercaptoethanol, and 1 ug/mL ILL 9 mer peptide.

### 2.8. ^51^Cr Release Cytotoxicity Assay 

ILL-stimulated splenocytes were co-incubated with ^51^Cr-labeled [19] T2 target cells in 96-well round-bottom plates at effector: target ratios of 100:1 (500,000:5000), 50:1 (250,000:5000), 25:1 (125,000:5000), and 12.5:1 (62,500:5000)). For ^51^Cr labeling, 2 × 10^6^ T2 cells were incubated with 1.85 MBq of ^51^Cr and 50 µg/mL of ILL 9 mer peptide for 1 h at 37 °C. Following 4 h at 37 °C, 50 µL of the supernatant of each well was transferred to Luma plates (PerkinElmer LAS (UK) Limited, Llantrisant, UK) and left to dry until reading on a Top Count microplate scintillation counter. Target cells alone incubated in medium constituted the spontaneous release value and target cells with medium + 10 µL of 10% *v*/*w* Sodium Dodecyl Sulfate (SDS) constituted the maximum release value. Cytotoxicity was calculated according to the following formula:$$\frac{\left(\mathrm{Experimental}\;\mathrm{release}-\mathrm{spontaneous}\;\mathrm{release}\right)\times 100=\cdots \%\;\mathrm{of}\;\mathrm{Cytotoxicity}}{\left(\mathrm{Maximum}\;\mathrm{release}-\mathrm{spontaneous}\;\mathrm{release}\right)}$$

### 2.9. Flow Cytometry-Based Cytotoxicity Assay

CD8^+^ T cell cytotoxicity was determined using a flow cytometry-based assay. For this, T cells were isolated by negative selection from ILL-stimulated splenocytes using the Dynabeads™ Untouched™ Mouse CD8 Cells Kit (Invitrogen, ThermoFisher Scientific, Paisley, UK). Murine splenic CD8^+^ T cells or human PBMCs were then co-incubated with LNCaP targets that had been fluorescently labeled with PKH26 (Sigma-Aldrich, Merck Life Science UK Limited, Gillingham, UK) as per the manufacturer’s recommended protocol in 12 × 75 mm polycarbonate tubes at effector: target ratios of 20:1 (400,000:20,000), 5:1 (100,000:20,000), and 1:1 (20,000:20,000). Following 3 h at 37 °C, cells were stained with LIVE/DEAD™ Fixable Yellow Dead Cell Stain (Invitrogen, ThermoFisher Scientific, Paisley, UK ) for 30 min in the dark. Samples were acquired and anlysed as described in Section 2.3. Target cells alone constituted the spontaneous percentage of dead target cells value. Cytotoxicity was calculated according to the following formula:$$\frac{\left(\%\;\mathrm{dead}\;\mathrm{target}\;\mathrm{cells}-\%\;\mathrm{spontaneous}\;\mathrm{dead}\;\mathrm{target}\;\mathrm{cells}\right)\times 100=\%\;\mathrm{of}\;\mathrm{Cytotoxicity}}{\left(100-\%\;\mathrm{spontaneous}\;\mathrm{dead}\;\mathrm{target}\;\mathrm{cells}\right)}$$

### 2.10. Ex Vivo Expansion of Human PBMCs

In order to increase the frequency of PAP-specific T cells, cryopreserved PBMCs were thawed in CTL thaw solution (RPMI 1640 medium containing 10% *v*/*v* CTL thaw solution (SLS/Lonza, UK) and 0.02% *v*/*v* Benzonase™ (Millipore, Merck Life Science UK Limited, Gillingham, UK). PBMCs (2 × 10^6^ cells/mL in a 24-well plate) were cultured in TexMACS™ serum-free medium (Miltenyi Biotec, UK) supplemented with 5% *v*/*v* human pooled serum (Sigma-Aldrich, Millipore, Merck Life Science UK Limited, Gillingham, UK) and stimulated with 1 µg/mL of ILL 9 mer peptide or 10 µg/mL of MutPAP42mer peptide for 4 days at 37 °C, after which recombinant human IL-2 (rhIL-2, 10 IU/mL) and rhIL-15 (10 ng/mL, R&D Systems, Biotechne, Abingdon, UK) were added and PBMCs incubated for a further 6 days at 37 °C. PBMCs were washed and resuspended in TexMACS™ medium containing 10 IU /mL rhIL-2 and incubated for a further 2 days at 37 °C. PBMCs were washed in TexMACS™ medium prior to Dextramer™ staining and cytotoxicity assays.

### 2.11. Dextramer™ Staining

Fc receptors on PBMCs or splenocytes (0.5–1 × 10^6^) were blocked by incubating with a human FcR blocking reagent (Miltenyi Biotec, Woking, UK, Catalogue number 130-059-901) or a murine CD16/CD32 mAb (BioLegend, London, UK clone 93) for 15 min at 4 °C, after which they were incubated with 16–32 × 10^−9^ M of the appropriate MHC class I Dextramer™ (Immudex, Copenhagen, Denmark) for 10 min at 4 °C in the dark. PBMCs were then incubated with human mAbs for CD8, CD3, and CD19 (Table A2) and splenocytes with murine mAbs for CD8, CD3, PD-1, Tim-3, and LAG-3 for 30 min at 4 °C in the dark. LIVE/DEAD™ Fixable Yellow Dead Cell Stain (Life Technologies, ThermoFisher Scientific, Paisley, UK) identified viable cells. Samples were acquired and analyzed as described in Section 2.3.

### 2.12. Statistical Analysis

Statistical analysis for all experiments was performed using GraphPad Prism 7 software. The *p*-values were calculated using either Student’s *t*-test with two-tailed distribution or two-way/one-way ANOVA, as stated. The *p*-values were annotated as follows: * *p* < 0.05, ** *p* < 0.01, *** *p* < 0.001, **** *p* < 0.0001.

## 3. Results

### 3.1. Pre-Existence of ILL-Specific CD8^+^ T Cells in the Blood of Patients with PCa

PAP-specific proliferative CD4^+^ ‘helper’ T cell responses were detected in 9 of 80 patients with PCa, irrespective of their disease stage [20]. In addition, while T cells specific for the ILL 9 mer HLA-A*02:01-restricted epitope have previously been described in HLA-A*02:01 healthy individuals and patients with PCa, these were not identified using a Dextramer™-based approach [21]. The presence of pre-existing immunity is important, as high pre-existing immunity towards the native HER-2 peptide, AE36, in patients with PCa has been shown to correlate with longer PFS following vaccination with a HER-2/neu hybrid peptide vaccine [22].

To better evaluate the potential efficacy of our new vaccine approach in patients, we used an HLA-A*02:01 ILL Dextramer™ (Immudex, Copenhagen, Denmark) to assess the prevalence of HLA-A*02:01 ILL-specific CD8^+^ T cells in PBMCs from a cohort of 16 patients with PCa, 5 individuals with diagnosed benign prostate disease, and 4 healthy individuals (Table A3). Two stimulating conditions to expand the ILL-specific CTL population were compared: using the ILL 9 mer minimal epitope or the MutPAP42mer peptide. No differences in the proportion of CD8^+^ T cells within the total T cell population in patients with PCa, individuals with benign prostate disease and healthy individuals were observed, nor were there any differences between stimulating conditions (Figure 1a). Stimulation with the ILL peptide was more efficient at expanding the population of ILL-CTLs than stimulation with the MutPAP42mer peptide (Figure 1b), a phenomena which was expected since CD8^+^ T-cell responses are best detected by restimulation with minimal epitopes, whereas the use of long peptide/protein tends to favor CD4^+^ T-cell responses.

Within the cohort of patients with PCa, ILL-specific CTLs following ILL peptide stimulation were detectable in six patients (37.5%, ranging from 0.29 to 1.6% of the CD8^+^ cells), whereas ILL-specific CTLs following MutPAP42mer peptide stimulation were only detectable in four patients (26.7%, ranging from 0.30 to 1.09% of the CD8^+^ T cell population) (Table A4). ILL-specific CTLs were detectable in two of the five individuals with benign disease, one after ILL peptide stimulation, and one after stimulation with the mutPAP42mer (0.92% and 0.5% of the CD8^+^ T cell population, respectively). Overall, three patients (LE109, 111 and 312) exhibited detectable ILL-specific CTLs only following ILL peptide stimulation and one patient (LE103) had detectable ILL-specific CTLs only following MutPAP42mer peptide stimulation. ILL-specific CTLs were not detectable in any of the healthy individuals tested (Table A4). This correlates with previous studies from other groups describing the presence PAP 135-143 (ILL)-specific T cells in 40% of patients with PCa (6 of 15) [23]. Studies have also demonstrated the presence of PAP-specific T cells that could secrete IFNγ in response to in vitro stimulation with PAP-derived peptides and/or lyse peptide pulsed T2 cells in healthy individuals (Stanford University Blood Center) [24], as well as in patients with PCa (University of Washington Medical Center and at the University of Wisconsin) [23]. Hadaschik et al. found a correlation between the proportion of ILL-CTLs (characterized as ILL-specific IFNγ-secreting CTLs) and the stage of the disease, with a higher proportion of patients with mCRPC displaying ILL-CTLs in comparison to patients with low-, intermediate-, or high-risk PCa [25]. In the current study, no correlation was found between the presence of circulating ILL-CTLs and the stage of the disease (low, intermediate, or high risk) (data not shown), although only 1 patient out of 16 was known to have metastatic PCa.

To determine if ILL-specific CTLs could lyse PCa cells, we performed a flow cytometry-based cytotoxicity assay against wild type LNCaP cells and LNCaP cells, which had been knocked down for PAP gene expression using shRNA. These LNCaP-PAP-low cells exhibited an 84% reduction of PAP expression at the mRNA level (Figure 1c). Although not statistically significant, there was a higher cytotoxicity against WT LNCaP cells in comparison to LNCaP PAP^low^ cells when using PBMCs stimulated with the ILL epitope (Figure 1c). This suggests that the ILL epitope is presented by HLA-A2 molecules at the surface of LNCaP cells. Comparison of the cytotoxicity against LNCaP cells following stimulation with the ILL epitope or with the mutPAP42mer peptide demonstrated that PBMCs stimulated with the mutPAP42mer sequence had a higher capacity to lyse LNcaP cells (Figure 1d). This might be due to CTLs recognising other HLA-A2 epitopes present in the mutPAP42mer peptide and exposed at the surface of LNCaP.

In conclusion, using ILL or mutPAP42mer antigen, PAP-specific T cells were detected in a large proportion of patients with PCa, thereby suggesting it to be a good therapeutic vaccine candidate. The presence of additional PAP42mer derived peptide-specific CTLs, which could not be assessed due to limited material, was assessed in the humanized double knock-out murine MHC transgenic mice HHDII/DR1.

### 3.2. The MutPAP42mer Vaccine Is More Immunogenic Than Its WT Counterpart

We have previously shown that a 15 mer PAP-derived sequence (underlined below in red font) administered as a DNA vaccine could induce strong vaccine-specific T cell responses in both C57Bl/6 mice and HHDII/DR1 transgenic mice and reduce tumor growth in a syngeneic heterotopic murine model of PCa [2]. We have subsequently elongated the sequence to 42 amino acids long (highlighted in red font in the sequence below) to include additional epitopes, and replaced alanine at position 116 (in blue) to a leucine in order to increase the predicted binding of 2 peptides (Table 2).

#### Full Human PAP Protein Sequence

MRAAPLLLARAASLSLGFLFLLFFWLDRSVLAKELKFVTLVFRHGDRSPIDTFPTDPIKESSWPQGFGQLTQLGMEQHYELGEYIRKRYRKFLNESYKHEQVYIRSTDVDRTL**MSAMTNLAALFPPEG**VSIWNPILLWQPIPVHTVPLSEDQLLYLPFRNCPRFQELESETLKSEEFQKRLHPYKDFIATLGKLSGLHGQDLFGIWSKVYDPLYCESVHNFTLPSWATEDTMTKLRELSELSLLSLYGIHKQKEKSRLQGGVLVNEILNHMKRATQIPSYKKLIMYSAHDTTVSGLQMALDVYNGLLPPYASCHLTELYFEKGEYFVEMYYRNETQHEPYPLMLPGCSPSCPLERFAELVGPVIPQDWSTECMTTNSHQGTEDST.

Herein, we assessed the immunogenicity of these 42 mer PAP-derived sequences in HHDII/DR1 transgenic mice. For this, male HHDII/DR1 mice were immunized with either the wild type (WT) or the MutPAP42mer peptide using CpG ODN 1826 as adjuvant. The number of IFNγ-releasing splenocytes in response to stimulation with HLA-A*02:01 and HLA-DRB1*01:01 epitopes contained within the WT PAP42mer sequence (Table A2) was measured using an IFNγ ELISpot assay. The MutPAP42 vaccine induced higher numbers of PAP-specific IFNγ-releasing splenocytes compared to wtPAP42 (Figure 2A). The response to the ILL HLA-A*02:01 epitope was significantly greater (90 IFNγ-releasing splenocytes per 0.5 × 10^6^ cells for wtPAP42 to 200 for mutPAP42, *p* < 0.001). This epitope is specifically of interest in the context of PCa, as CD8^+^ T cells specific for this epitope have been identified in the blood of patients and in the PBMCs of healthy donors after multiple in vitro stimulations [23].

Functional avidity of a T cell receptor (TCR) is defined by the T cell responsiveness towards an epitope and depends, partly, on its affinity. T cells with high functional avidity respond to low quantities of peptides [26] nd are required to establish an effective anti-tumor immune response [27]. Therefore, we measured the functional avidity of splenocytes against the ILL HLA-A*02:01 epitope by performing the IFNγ ELISpot assay in the presence of decreasing concentrations of peptide. We observed high numbers of IFNγ-releasing splenocytes when stimulated with low concentrations of ILL only within the MutPAP42mer vaccine group (Figure 2B top). This is further verified by the 10-fold higher calculated EC50 (0.01267 μg/mL) for the WT vaccine in comparison to the mutated vaccine (0.001049 μg/mL) (Figure 2B bottom). Interestingly, despite being unaffected by the mutation, and thus, in the absence of difference in the predicted binding score, the immunogenicity of the ILL epitope was highly elevated due to the mutation. Both the ALF and the SAM epitopes, the latter of which was predicted to have an improved binding capacity as a direct result of the mutation (original binding score 24 to 30 with the mutation), also showed improved immunogenicity, although not to the same extent as the ILL. It is, therefore, possible that the mutation affected the endogenous processing of the 42 mer, maintaining ILL as the immuno-dominant peptide over the other peptides, irrespective of binding capacity. Taken together, these data demonstrate the capacity of the MutPAP42mer vaccine to induce PAP-specific splenic T cells in the context of HLA-A*02:01 and of HLA-DRB1*01:01.

### 3.3. CAF^®^09 Adjuvant Increases the Immunogenicity of the Vaccine

The ability of the CAF^®^09 adjuvant to further improve the immunogenicity of the MutPAP42mer vaccine was then tested. The number of IFNγ-releasing splenocytes in response to ILL HLA-A*02:01 epitope stimulation was seven-fold higher when using CAF^®^09 compared to CpG (from 50 to 350, *p* < 0.0001) (Figure 3A). An identical IFNγ response was observed against the VSIW HLA-DRB1*01:01 epitope, which contains the ILL HLA-A*02:01 epitope within its sequence. CpG was, however, found to increase the T cell responses towards a broader repertoire of the peptides tested. This might be related to a better compatibility of the ILL and VSIW peptides together with CAF^®^09, e.g., stronger binding than the other peptides leading to stronger presentation of this single peptide at the expense of the others, thus imposing a dominance hieracy between the peptides.

Despite inducing more IFNγ-releasing splenocytes, the CAF^®^09-based vaccine did not improve T cell avidity towards the ILL HLA-A*02:01 epitope (EC50 0.001049 μg/mL for CpG; 0.001435 μg/mL for CAF^®^09) (Figure 3B). These data demonstrate that both CAF^®^09 adjuvant and CpG induce high numbers of IFNγ producing T-cells.

### 3.4. Vaccine-Induced Immune Response towards PAP Antigen Is Driven by CD8^+^ T Cells

To further characterize the immune response generated by the MutPAP42mer vaccine, splenic T cells were immunophenotyped by flow cytometry [28]. The proportion of CD8^+^ T cells within the total in animals immunized with the CAF^®^09 vaccine in conjunction with CAF^®^09 (15%) was higher than that in non-immunized (naïve) animals and animals immunized with the vaccine in conjunction with CpG (5% and 8% respectively, *p* < 0.0001) (Figure 4a). Immunization using CAF^®^09 led to a three-fold increase in CD44^high^CD62L^-^ effector/effector memory CD8^+^ T cells (20 to 60%, *p* < 0.0001), as defined as CD44^high^CD62L^-^ by (Sckisel et al., 2017), whereas vaccination using CpG had no effect on the prevalence of these cells (Figure 4b). The percentage of effector/effector memory CD4^+^ T cells was not affected by the vaccination (Figure 4b), nor was any change in the proportion of CD44^high^CD62L^+^ central memory T cells observed (data not shown).

We also determined the expression of several inhibitory receptors on the surface of T cells. CTLA-4, LAG-3, Tim-3, and PD-1 receptors are transiently expressed following TCR activation [29,30,31,32] and bind to ligands expressed at the surface of tumor cells and of some immune cells (DCs, monocytes, B lymphocytes). Their expression is retained in case of antigen persistence and a sustained pro-inflammatory milieu in order to control auto- reactivity and maintain peripheral tolerance [33,34]. Indeed, prolonged expression of these receptors leads to the loss of functional capacities such as proliferation, secretion of pro-inflammatory cytokines (IFNγ, TNFα, and IL-2), and degranulation [35]. Inhibitory receptors have therefore been extensively studied as markers of dysfunction/exhaustion of T cells in cancer and antibodies targeting them have been successful in improving immunity against several cancer types [36]. The expression of PD-1 by CD4^+^ (*p* < 0.001) and CD8^+^ (*p* < 0.0001) T cells following immunization using CAF^®^09 was increased, whereas no significant changes in the expression of CTLA-4, LAG-3, or Tim-3 were observed (Figure 4c).

To confirm PAP and HLA-A*02:01 specificity of the immune response, the functional responsiveness of vaccine-induced T cells was assessed by measuring the release of cytokines, and degranulation and proliferation of T cells after stimulation with the ILL HLA-A*02:01 epitope alone or with the VSIW HLA-DR1 epitope for 6 h. The CAF^®^09-based vaccine induced greater numbers of ILL-specific CD8^+^ T cells than the CpG-based vaccine (Figure 4d). Indeed, 7% of CD8^+^ T cells secreted IFNγ and TNFα (*p* < 0.01 versus CpG, *p* < 0.0001 against no peptide control), with a non-significant increase in IL-2 secreting CD8^+^ T cells. CD8^+^ T cells proliferated in response to stimulation with the ILL epitope, as demonstrated by an increased expression of Ki67 (*p* < 0.001). Finally, the cytotoxic potential of T cells was assessed by measuring the expression of CD107a, a marker of degranulation, and the cytolytic granule Granzyme B. The proportion of CD8^+^ T cells expressing CD107a and Granzyme B was increased (*p* < 0.01, non-statistically significant, respectively). Interestingly, the increased proportion of CD8^+^ T cells expressing Granzyme B was not dependent on ex vivo stimulation with ILL. The secretion of IFNγ and TNFα was restricted to CD8^+^ T cells, as stimulation with the VSIW HLA-DRB1*01:01 epitope did not induce a CD4^+^ T cell response (Figure 4d bottom left). However, it is possible that the assay performed as described may lack the necessary pre-stimulation time before the addition of monensin/brefeldin for CD4^+^ T cell to produce IFNγ [37]. Taken together, these data demonstrate the induction of functional PAP-specific CD8^+^ T cells following immunization with the MutPAP42mer/CAF^®^09 vaccine.

### 3.5. Vaccination Induces Secretory and Cytotoxic ILL-Specific Cytotoxic T Lymphocytes (CTLs) Co-Expressing PD1, TIM3, and LAG3

The presence of ILL-specific CD8^+^ cytotoxic T lymphocytes (CTLs) was then assessed by performing a cytotoxicity assay against ILL-presenting target cells. For this, splenocytes were stimulated ex vivo with ILL epitope and recombinant murine IL-2 for 6 days. Although the CTL expansion protocol increased the proportion of CD8^+^ T cells in all groups compared to freshly isolated splenocytes (Figure 4a versus Figure 5a), the proportion of CD8^+^ T cells after vaccination with the CAF^®^09-based vaccine was 2.7-fold and 2-fold greater than that in naïve animals and animals vaccinated with the CpG-based vaccine, respectively (*p* < 0.0001) (Figure 5a).

Interestingly, CD8^+^ T cells from all groups expressed LAG-3 and Tim-3, but no CTLA-4 (Figure 4b). PD-1 expression was vaccine-dependent, with 30% of CD8^+^ T cells in the naïve group expressing PD-1, 70% in the CpG vaccine group and 90% in the CAF^®^09-based vaccine group (*p* < 0.0001) (Figure 5b). Despite the co-expression of PD-1, Tim-3, and LAG-3, CD8^+^ T cells from vaccinated animals retained the capacity to secrete IFNγ and TNFα and to degranulate in an HLA-A*02:01 ILL epitope-specific manner, as demonstrated by the expression of CD107a and Granzyme B (*p* < 0.0001) (Figure 4c).

Moreover, ILL-specific CTLs generated with both the CpG- and the CAF09^®^-based vaccine lysed ILL-pulsed T2 target cells in a ^51^Cr release cytotoxicity assay (*p* < 0.0001) (Figure 5d). The CAF^®^09-adjuvant-based vaccine induced higher levels of cytotoxicity against ILL-pulsed T2 cells at all ratios tested (*p* < 0.0001).

However, it should be noted that the proportion of CD8^+^ T cells in the splenocytes from mice immunized using the CAF^®^09-based vaccine was twice that of splenocytes from animals immunized using the CpG-based vaccine. Interestingly, the correlation between the proportion of CD8^+^ T cells co-expressing PD-1, Tim-3, and LAG-3 and the level of cytotoxicity towards ILL-presenting T2 cells confirmed that, in this model, these markers do not characterize dysfunctional CTLs; rather on the contrary, i.e., highly functional CTLs with cytotoxic functions (Figure 5e).

Finally, the cytotoxicity against LNCaP cells naturally expressing the PAP antigen was assessed by flow cytometry. The use of total splenocytes as effector cells led to inconclusive results (data not shown). Therefore, CD8^+^ T cells isolated from total splenocytes were used as effector cells. CD8^+^ T cells from vaccinated animals induced a higher level of cytotoxicity against LNCaP cells in comparison to those from naïve animals, with the highest comparative cytotoxicity at the effector:target ratio of 20:1 being significant (Figure 5f).

### 3.6. HLA-A*02:01 ILL-Specific CTLs Express PD-1

The specificity of vaccine-induced CTLs towards HLA-A*02:01 ILL epitope was also evaluated by flow cytometry using an HLA-A*02:01 ILL Dextramer™ (Immudex). Following vaccination with the MutPAP42mer peptide in the context of CAF^®^09, 1.5% of the total CD8^+^ T cells expressed a TCR binding specifically to ILL/HLA-A*02:01 complexes (*p* < 0.01) (Figure 6A). Moreover, most of these cells (85%) expressed PD-1 (Figure 6B). After ex vivo expansion of ILL-specific CTLs, 40% of the total CD8^+^ T cells from vaccinated animals expressed a TCR binding specifically to ILL/HLA-A*02:01 complexes (*p* < 0.001 versus Dextramer™ control and *p* < 0.05 versus naïve animals) (Figure 6C). In these cultures, 95% of the cells co-expressed PD-1, Tim-3, and LAG-3 (Figure 6D). In other words, all ILL-Dextramer™ positive CTLs express PD-1, but not all PD-1^+^ CTLs are ILL-specific CTLs. Approximately 50% of PD-1^+^ CD8^+^ T cells after ex vivo expansion were ILL-specific CTLs detectable by Dextramer™, which corresponds to the proportion of CD8^+^ T cells secreting IFNγ and TNFα following ILL stimulation (Figure 5c).

## 4. Discussion

Our study focused on the use of PAP protein as a basis for the development of a therapeutic vaccine against PCa. PAP is an attractive target due to its relative, prostate-specific expression and its disease-dependent overexpression. Moreover, the FDA approval of the PROVENGE^®^ (sipuleucel-T) vaccine in 2010 demonstrated the rationale for using PAP as a lead candidate for a PCa vaccine. However, vaccines using the entire PAP protein or DNA coding for the entire protein might increase not only effector T cells against PAP, but also potentially PAP-specific immunoregulatory T (Treg) cell populations. Moreover, when HIV whole protein was compared with HIV-derived overlapping 24-mer long peptide, Zhang et al. found that, due to a difference in trafficking, only 24-mer peptides could activate both CD8^+^ and CD4^+^ T cells, whereas whole protein, which was not found in the cytosol, but only in the endosomal compartment, could only activate CD4^+^ T cells [8].

Our study was performed in the context of HLA-A2 and DR1 haplotypes. The HLA-A2 haplotype is common in all ethnicities, with an average of 47.6% of the population being HLA-A2^+^ [37]. Interestingly, the HLA-A2 haplotype in patients with PCa seems over-representated (73%) compared to healthy individuals (58%), which might suggest a negative prognostic correlation with the HLA-A2 haplotype [38]. However, PAP is a ‘self-antigen’ and tolerance towards it poses a challenge for its efficacy as a vaccine antigen. Notwithstanding this, a number of clinical studies have demonstrated the possibility of breaking tolerance using xenoantigens (e.g., gp100, tyrosinase, and PAP) without compromising safety [39,40].

In order to overcome tolerance towards PAP, while limiting the chance of generating PAP-specific Treg cells, we introduced a mutation in the sequence of our vaccine, which increased immunogenicity towards wild type PAP-derived epitopes.

In a cohort of 16 patients with PCa, we found that 37.5% individuals had pre-existing ILL-specific CTLs in their blood and that their PBMCs could lyse LNCaP cells. However, more experiments are required to determine whether the presence of circulating ILL-CTLs increases in patients with PCa. Moreover, the phenotype and functional capacities of PAP-specific T cells in patients with PCa should be further characterized to evaluate which combinatorial immunotherapeutic treatment could provide the most benefit to our vaccine strategy.

Whether pre-existing immunity towards an HLA class I epitope can predict an improved overall survival (OS) in vaccinated patients is not clear. Voutsas et al. assessed a therapeutic peptide-based vaccine targeting the oncogene HER-2 in a phase I clinical trial for the treatment of patients with PCa [22]. Pre-existence of reactivity to two PSA-derived epitopes at high frequencies was detected and a further enhancement was observed after vaccination. However, although pre-existing immunity towards the HLA-A2 restricted epitope correlated with longer PFS, high pre-existing and vaccine-induced immunity towards the HLA-A2 restricted epitope showed a trend towards a shorter PFS.

In the context of myeloid malignancies, the percentage of HLA-A2-restricted epitope tetramer positive CTLs before vaccination has not been shown to associate with immune responders after vaccination. However, a greater increase in avidity of TCRs on epitope-specific CTLs following vaccination has been observed in clinical responders, thereby demonstrating the importance of measuring the quality of epitope-specific CTLs, rather than just their numbers [41].

In the context of the pTVG-HP PAP DNA vaccine, two predictive parameters of the development of vaccine-induced PAP-specific effector responses have been described. Johnson et al. investigated baseline immune parameters that predicted the establishment of a PAP-specific immune response following vaccination with the pTVG-HP PAP DNA vaccine [42]. Although both responders and non-responders exhibited PAP-specific Th1 responses before vaccination, non-responders displayed a higher PAP-specific secretion of IL-10 (*p* = 0.09) by both CD4^+^ and CD8^+^ T cells. Therefore, it appears that pre-existing regulatory-type antigen-specific T cell immunity is a negative predictive parameter of responsiveness to a PAP DNA vaccine. Another potential negative predictive marker for the development of PAP-specific immunity after vaccination is the presence PAP-specific CD8^+^ regulatory T cells expressing CTLA-4 and secreting IL-35 [43]. Unlike IL-10 or TGF-β blockade, CTLA-4 and IL-35 blockade could reverse their suppressive phenotype. However, none of these studies correlated the presence of the negative predictive markers with OS.

In addition to assessing the presence of antigen-specific T cells prior to vaccination, Farsaci et al. have calculated a “peripheral immunoscore” which can predict the OS benefit of patients with PCa before receiving the PROSTVAC^®^ vaccine, which is deduced by the presence of specific immune cell subsets prior to vaccination [44]. High frequencies of PD-1^+^ and CTLA-4^+^ CD4^+^ T cells and of differentiated CD8^+^ T cells, along with low frequencies of Treg cells and differentiated CD4^+^ T cells have been shown to correlate with improved OS following treatment with the GVAX vaccine (GVAX vaccine is a cellular vaccine consisting of two human PCa cell lines, LNCaP (CG8711) and PC-3 (CG1940), which have been transduced with an adeno-associated viral vector to secrete GM-CSF) in combination with Ipilimumab [45]. Moreover, high frequencies of myeloid-derived suppressor cells (MDSCs) before treatment were associated with shorter OS [46]. Taken together, these studies demonstrate the feasibility and utility of pre-vaccine immune profiling for predicting the efficacy of a therapeutic vaccine for PCa and enabling the selection of individuals that are most likely to benefit.

In order to overcome tolerance towards PAP, we introduced a mutation in the sequence of our vaccine, which was shown to increase the immunogenicity towards wild type PAP-derived epitopes. Moreover, the immunogenicity of the vaccine was further enhanced by combining it with the CAF^®^09 adjuvant. Our results demonstrated the induction of HLA-A2 epitope-specific IFNγ producing CD8^+^ T cells following vaccination. The CAF^®^09 adjuvant demonstrated IFNγ restricting the CD8 T cell recognition to the PAP135-143 ILL HLA-A*02:01-restricted epitope in the HHDII/DR1-humanized mouse model. The observation that CAF^®^09 selectively enhances the responses against certain epitopes at the expense of others imposing an epitope dominance hierarchy suggests that these peptides employ properties that makes them a better match for CAF^®^09. Such properties could be better association with the adjuvant or intracellular desorption better facilitating cross-presentation. Future studies could, thus, encompass studies on how to broaden the epitope repertoire, e.g., by modifying the physical chemical properties of the peptides or by differentiating the concentrations of the antigens.

CTLs specific for this epitope have previously been reported in the blood of healthy individuals and patients with PCa [21,23,25] and these cells have been shown to lyse ILL-presenting target cells and LNCaP cells in vitro [21]. Our vaccine induced high avidity, functional ILL-specific CTLs that could lyse ILL-presenting cells in vitro. Although we did not observe the induction of a CD4^+^ T cell responses in our model using intracellular flow cytometry, we did observe IFNγ responses against the CD4 epitopes which are indicative of a Th1 response in the ELISpot assay, a finding which others have also described following vaccination with CAF^®^09 adjuvant [15,47]. CD4^+^ T cell help is crucial for the induction of CD8^+^ memory T cells and priming in the presence of CD4^+^ helper T cells can also increase the expression of cytotoxic effector molecules and decrease the expression of inhibitory receptors such as PD-1 and LAG-3 [48]. The absence of a pool of central memory CD8^+^ T cells as assessed by their expression of CD44^high^CD62L^+^ [28] in our model might be due to the timepoint of spleen harvesting (one week following the last immunization), as others have described the progressive loss of effector memory T cell pools over time and subsequent establishment of a stable pool of central memory T cells [49]. It might, therefore, be that central memory T cells would be observed when harvesting spleens 3–4 weeks after the final immunization.

Although T cells co-expressing PD-1, Tim-3, and LAG-3 receptors have been referred to as ‘exhausted’ [32,50], we have found that vaccine-induced T cells co-expressing these receptors remained functional, as demonstrated by the capacity to degranulate, to secrete IFNγ and TNFα and to lyse ILL-presenting target cells. Others have also observed CTL effector function to be retained in T cells co-expressing inhibitory markers, such as the co-expression of LAG-3 and 4-1BB, which has been reported to characterize dysfunctional antigen-specific CD8^+^ T cells that are deficient in TNFα production, but retain IFNγ production in vitro. These findings suggest that more studies are needed to link the phenotype to the functional capacities of antigen-specific T cells.

Immune checkpoint blockade strategies which demonstrated efficacy in subsets of patients with PCa treated with Ipilimumab in combination with Sipuleucel-T and PSA-TRICOM vaccines showed encouraging results in clinical trials [51,52,53]. Preliminary results from combining Pembrolizumab with pTVG-HP DNA vaccine (targeting PAP) have reported reduced PSA levels and tumor volumes in several patients [54].

## 5. Conclusions

We have developed a vaccine strategy, potentially more cost-effective than PAP-whole protein targeted vaccines, that induces PAP-specific immune responses in a humanized mouse model. In vivo cytotoxic functions of vaccine-induced T cells is currently being assessed in order to evaluate the therapeutic potential of this vaccine for patients with PCa.

## Figures and Tables

**Figure 1 cancers-14-01970-f001:**
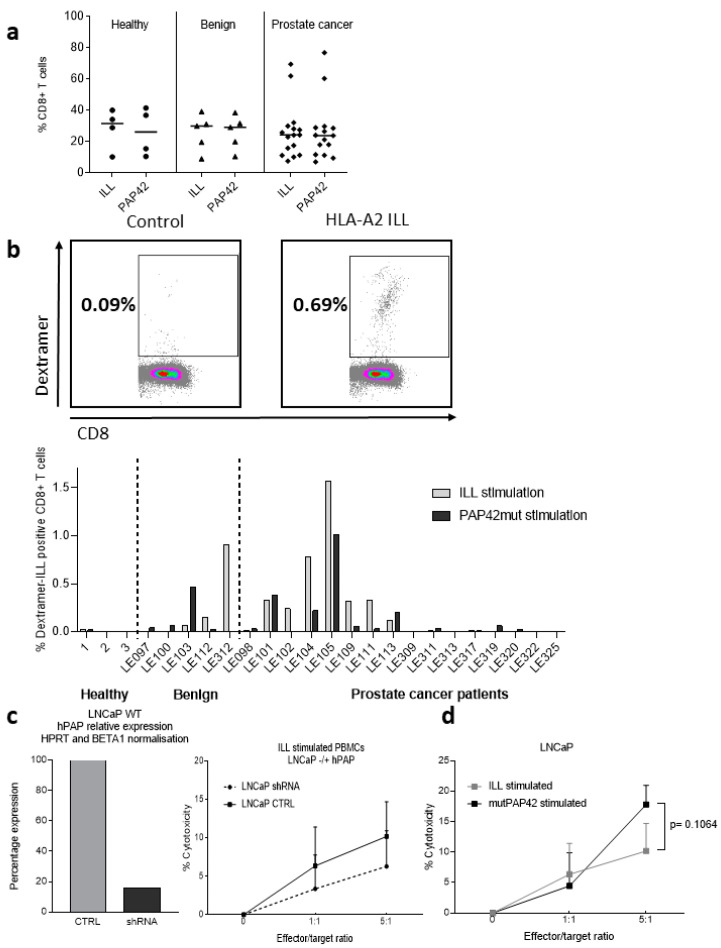
Presence of ILL-specific CD8^+^ T cells in the blood of patients with PCa. Cryopreserved PBMCs from HLA-A2^+^ individuals were cultured for 12 days in the presence of IL-2, IL-15, and either the ILL9mer peptide or the hPAP42mer-mutated peptide. Three groups were compared: healthy individuals, individuals with benign disease, and patients with PCa. On day 12, PBMCs were blocked with a human FcR block and then incubated with surface mAbs and the relevant Dextramer™. (**a**) Proportion of CD8^+^ T cells within total T cells, (**b**) gating example of Dextramer™-positive CD8^+^ T cells and proportion of Dextramer™-positive CD8^+^ T cells following each stimulating condition. (**c**) Cytotoxicity of ILL-stimulated PBMCs against PKH26-labeled WT or PAP^low^ LNCaP cells was measured by flow cytometry. (**d**) Cytotoxicity against PKH26-labeled WT LNCaP cells was compared between each stimulating condition. No difference in the proportion of CD8^+^ T cells was observed between groups or stimulating conditions. ILL-specific CD8^+^ T cells were detected following ILL9mer peptide stimulation (benign:1/5, PCa: 6/16) and after hPAP42mer-mutated peptide stimulation (benign: 1/5, PCa: 4/16). Cytotoxicity against LNCaP cells was observed. Bars represent the mean percentage of positive cells and the errors bars represent the SD (*n* = 3 to 16 patients per group). Dots represent the mean percentage of cytotoxicity and the errors bars represent the SED (**c**). Differences in the proportion of positive cells between groups were determined using a two-way ANOVA comparison test.

**Figure 2 cancers-14-01970-f002:**
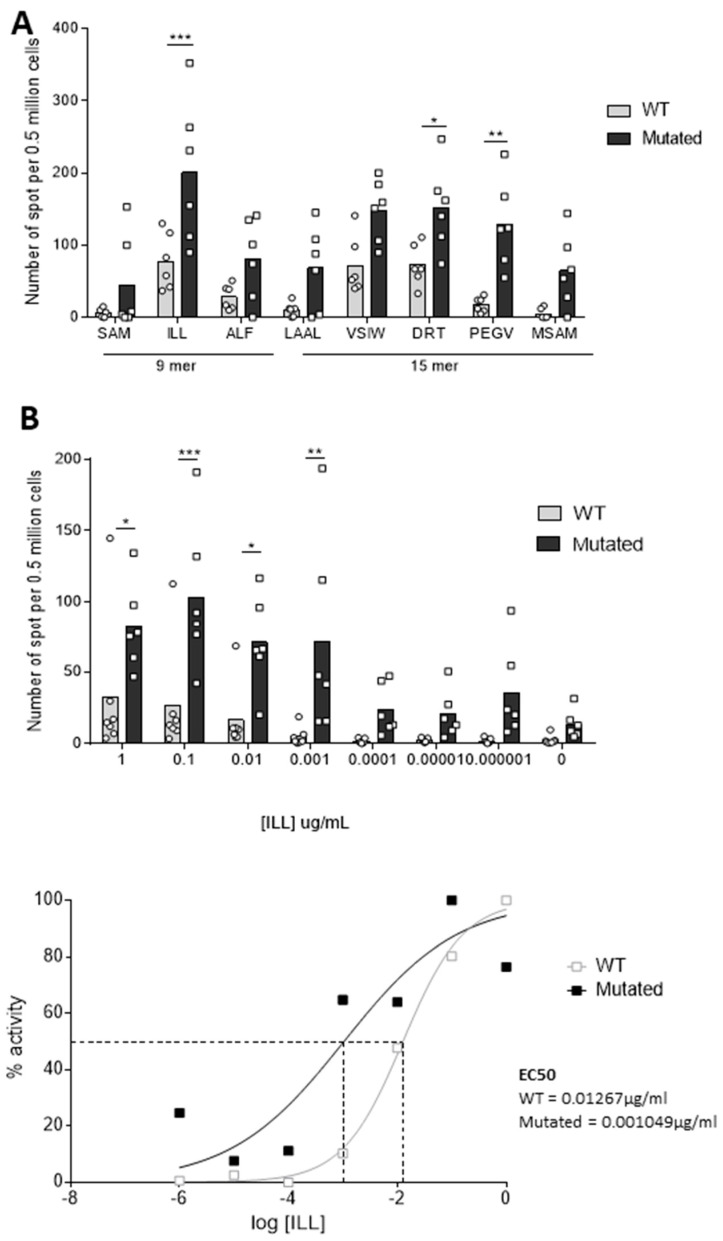
The mutated hPAP42mer peptide increases the immunogenicity of the vaccine against several class-I and class-II PAP-derived epitopes and increases the functional avidity towards the ILL HLA-A*0201 epitope. HHDII/DR1 mice were immunized on days 1, 15, and 29 with either hPAP 42 mer WT or mutated peptide with CpG adjuvant. Seven days after the last immunization, splenocytes were isolated from spleens and an in vitro IFNγ ELISpot assay was performed. Splenocytes were stimulated with (**A**) hPAP-derived class I and class II peptides or (**B**) decreasing concentrations of ILL 9 mer peptide, for 48 h at 37 °C. Immunization with the mutated hPAP42mer peptide induced higher numbers of peptide-specific IFNγ-releasing T cells and induced T cells with a higher functional avidity for ILL 9 mer peptide than immunization with the WT hPAP42mer peptide. Bars represent the mean number of spots, and the error bars represent the SD. Sigmoidal curve representing the functional avidity of ILL 9 mer peptide (**B**, bottom). Two independent experiments performed (*n* = 6–7 mice per test group). A significant difference in the induction of peptide-specific IFNγ-releasing T cells between immunization groups was determined using a two-way ANOVA comparison test. The *p*-values were annotated as follows: * *p* < 0.05, ** *p* < 0.01, *** *p* < 0.001.

**Figure 3 cancers-14-01970-f003:**
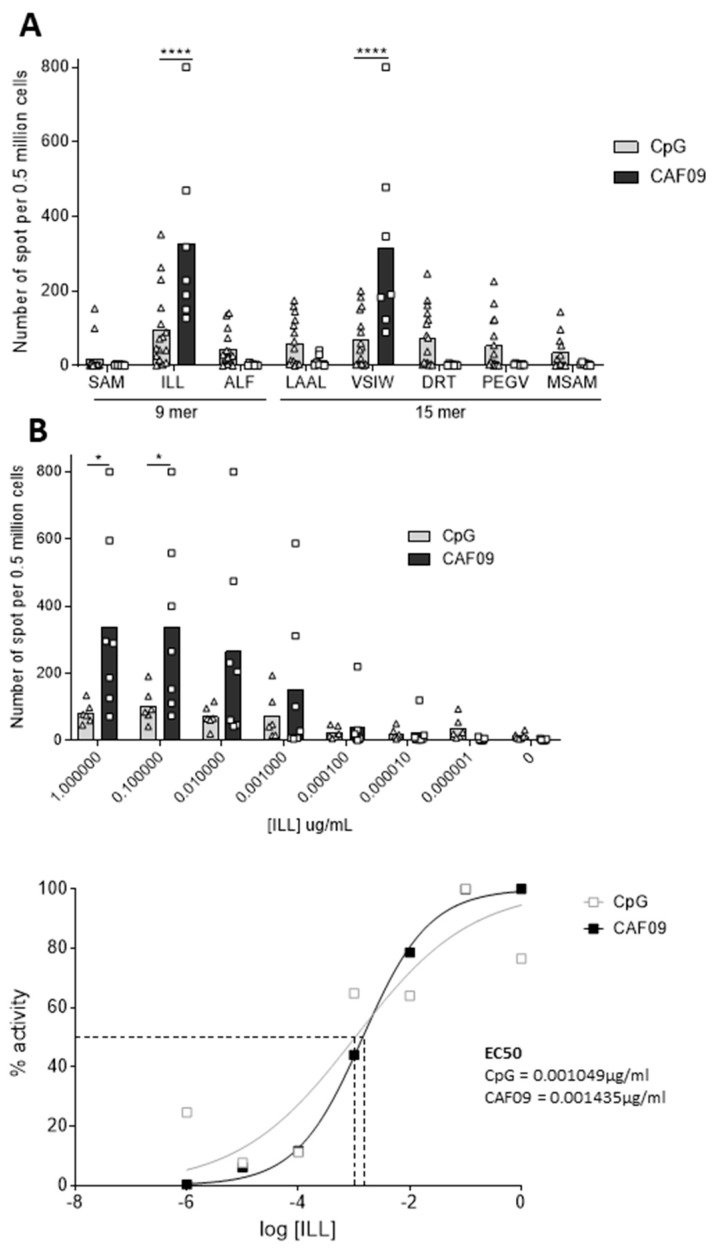
CAF^®^09 adjuvant increases the immunogenicity of the vaccine against ILL 9 mer and VSIW 15 mer epitopes, but does not increase the functional avidity towards ILL 9 mer epitope. HHDII/DR1 mice were immunized on days 1, 15, and 29 with either the CpG or the CAF^®^09-based mutated hPAP 42 mer vaccine. Seven days after the last immunization, splenocytes were isolated from spleens and an in vitro IFNγ ELISpot assay was performed. Splenocytes were stimulated with (**A**) hPAP-derived class I and class II peptides or (**B**) decreasing concentrations of ILL 9 mer peptide, for 48 h at 37°C. Immunization with the CAF^®^09-based vaccine induced higher numbers of peptide-specific IFNγ-secreting T cells than immunization with other vaccines. Immunization with CpG adjuvant-induced T cells with a higher functional avidity for ILL 9 mer peptide than immunization with the CAF^®^09 adjuvant. Bars represent the mean number of spots and the error bars represent the SD. Sigmoidal curve representing the functional avidity of ILL 9 mer peptide (**B**, bottom). (**A**) Four to six independent experiments performed (*n*= 13 to 20 mice per test group) and (**B**) one to two independent experiments (*n* = 3 to 6 animals per test group). A significant difference in the induction of peptide-specific IFNγ-releasing T-cells between immunization groups was determined using a two-way ANOVA comparison test. The *p*-values were annotated as follows: * *p* < 0.05, **** *p* < 0.0001.

**Figure 4 cancers-14-01970-f004:**
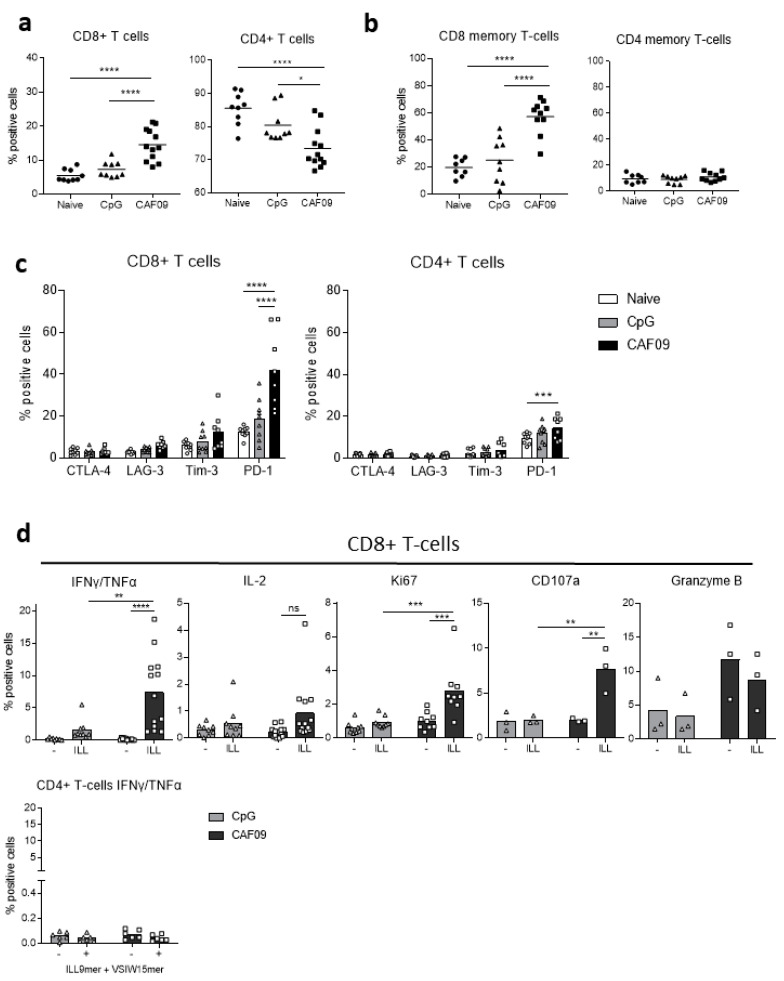
The PAP42mer-mutated vaccine promotes the expansion of effector/memory CD8^+^ Tcells and induces ILL-specific CD8^+^ T cells with a secretory and degranulating phenotype. HHDII/DR1 mice were immunized on days 1, 15, and 29 with either the CpG or the CAF09^®^-based mutated hPAP 42 mer vaccine. Seven days after the last immunization, splenocytes were isolated from spleens and incubated with a murine FcR block and stained with surface Abs for flow cytometry analysis indicating (**a**) proportion of T cells, (**b**) proportion of effector memory T cells, and (**c**) expression of inhibitory receptors. Splenocytes were stimulated with either ILL 9 mer alone or with VSI 15 mer for 6 h at 37 °C, incubated with a murine FcR block, stained with surface mAbs, fixed, and permeabilized and stained with intracellular mAbs for flow cytometry analysis indicating (**d**) cytokine secretion, proliferation, and degranulation. The CAF09^®^-based vaccine increased the proportion of CD8^+^ T cells and of CD8^+^ memory T cells and induced the expression of PD-1. The vaccine also induced the proliferation, cytokine secretion, and degranulation of CD8^+^ T cells upon stimulation. Bars represent the mean percentage of positive cells and the error bars represent the SD. One to four independent experiments were performed (*n* = 3 to 12 animals per test group). A significant difference in the proportion of positive cells between immunization groups was determined using a two-way ANOVA comparison test. The *p*-values were annotated as follows: * *p* < 0.05, ** *p* < 0.01, *** *p* < 0.001, **** *p* < 0.0001.

**Figure 5 cancers-14-01970-f005:**
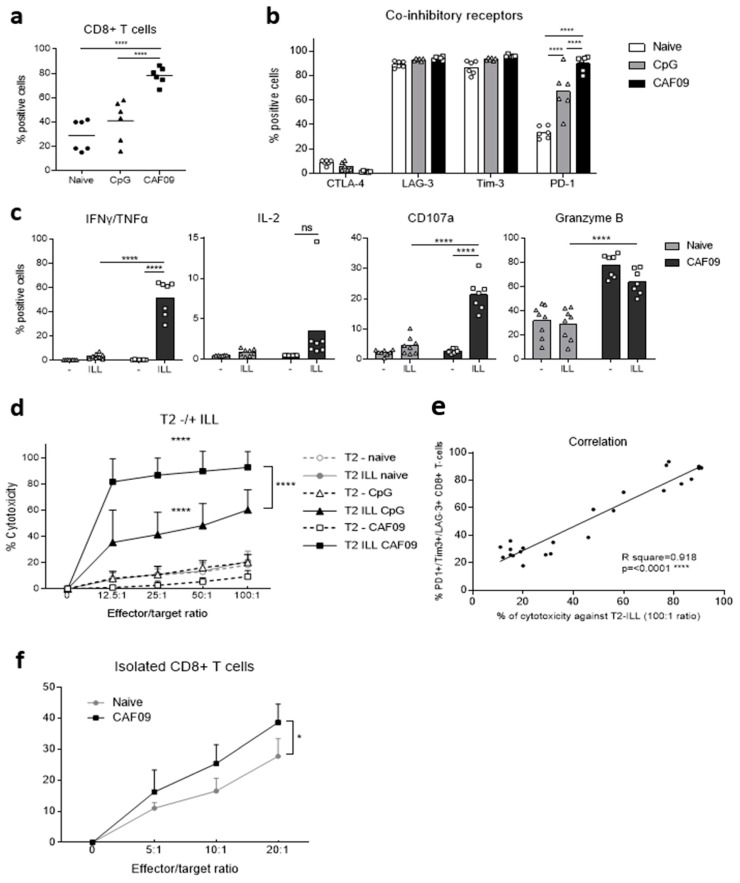
In vitro expanded ILL-specific CD8^+^ Tcells exhibit an exhausted phenotype while retaining highly secretory and cytotoxic function. HHDII/DR1 mice were immunized on days 1, 15, and 29 with either the CpG or the CAF09^®^-based mutated hPAP 42 mer vaccine. Seven days after the last immunization, splenocytes were isolated from spleens and incubated with ILL 9 mer peptide for 6 days, followed with 1 day of rest. Splenocytes were then incubated with a murine FcR block and stained with surface mAbs for flow cytometry analysis indicating (**a**) proportion of CD8^+^ T cells, (**b**) expression of inhibitory receptors, and (**c**) cytokine secretion and degranulation, following fixation, permeabilization, and staining with intracellular mAbs. For cytotoxicity experiments, splenocytes were co-incubated with ^51^Cr labeled peptide-pulsed target cells for 4 h and the radioactivity was measured to determine the percentage of cytotoxicity. (**d**) Percentage of cytotoxicity against ILL-pulsed or un-pulsed T2 cells and (**e**) correlation between the proportion of CD8^+^ T cells co-expressing PD-1, Tim-3, and LAG-3 receptors and the percentage of cytotoxicity against ILL pulsed T2 cells at 100:1 ratio. Alternatively, a flow cytometry-based cytotoxicity assay was performed by co-incubating for 3 h isolated CD8^+^ T cells as effector cells and PKH26-labeled LNCaP cells as target cells (**f**). Bars represent the mean percentage of positive cells and the error bars represent the SD (**a**–**c**). Dots represent the mean percentage of cytotoxicity and the errors bars represent the SED (**d**,**f**). Results are representative of two to three independent experiments (*n* = 6 to 9 animals per test group). The CAF09^®^-based vaccine increased the proportion of CD8^+^ T cells (**a**) and the expression of PD-1 (**b**) and induced ILL-specific CTLs with secretory and cytotoxic functions (**c**) capable of lysing ILL-presenting target cells (**d**). The co-expression of inhibitory receptors correlated with the capacity to lyse ILL-presenting target cells (**e**). CAF09^®^-based vaccine increased the cytotoxicity of CD8^+^ T cells against LNCaP cells (**f**). A significant difference in the proportion of positive cells between immunization groups was determined using a two-way ANOVA comparison test. The *p*-values were annotated as follows: **** *p* < 0.0001.

**Figure 6 cancers-14-01970-f006:**
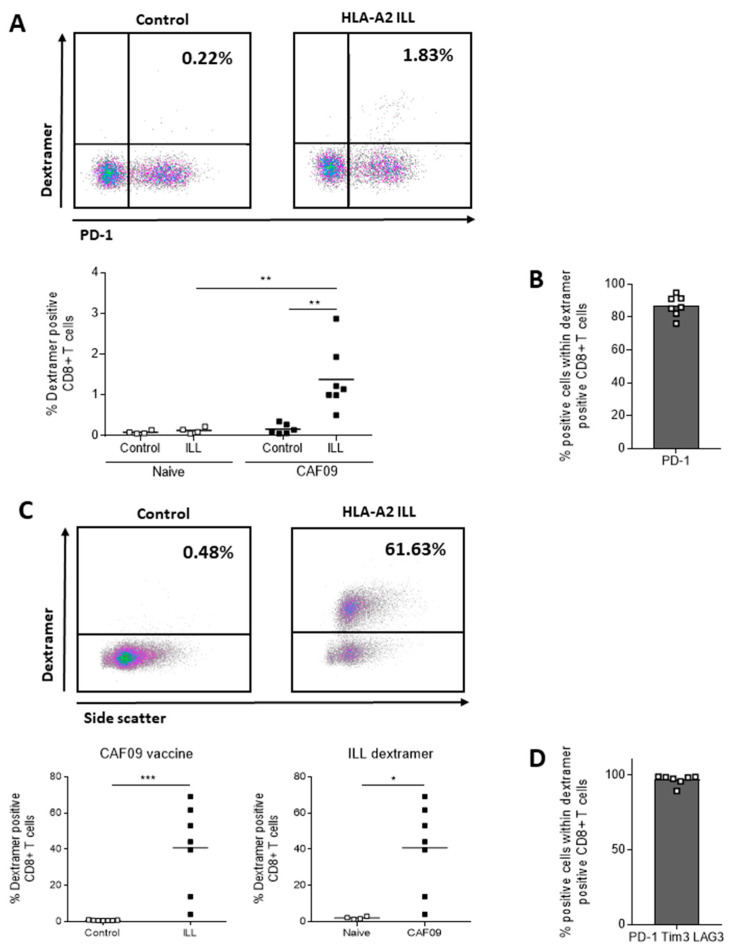
Vaccine-induced ILL-specific CD8^+^ T cells, as detected using Dextramer™ staining, almost exclusively express PD-1. HHDII/DR1 mice were immunized on days 1, 15, and 29 with either the CpG or the CAF^®^09-based mutated hPAP 42 mer vaccine. Seven days after the final immunization, splenocytes were isolated from spleens. Splenocytes were then incubated with a murine FcR block, stained with surface mAbs for flow cytometry analysis indicating (**A**) proportion of CD8^+^ T cells and (**B**) expression of inhibitory receptors, on freshly isolated splenocytes, or incubated with ILL 9 mer peptide for 6 days and stained for flow cytometry analysis indicating (**C**) proportion of CD8^+^ T cells and (**D**) expression of inhibitory receptors. CAF^®^09-based vaccine induced ILL-specific CD8^+^ T cells (**A**) expressing PD-1 (**B**), which were expanded with in vitro peptide stimulation (**C**) and co-expressed PD-1, Tim-3, and LAG-3 (**D**). The *p*-values were annotated as follows: * *p* < 0.05, ** *p* < 0.01, *** *p* < 0.001.

**Table 1 cancers-14-01970-t001:** Most recent clinical trials of PCa immunotherapeutics.

Status	Clinical Trials.gov Identifier	Interventions	Phase
Recruiting Date completion 2023	NCT04071236	Drug: Avelumab Drug: Peposertib Other: Quality-of-Life Assessment Other: Questionnaire Administration Radiation: Radium Ra 223 Dichloride	Phase 1 Phase 2
Completed	NCT00861614	Drug: Ipilimumab Drug: Placebo	Phase 3
Completed	NCT01057810	Drug: Ipilimumab Drug: Placebo	Phase 3
Recruiting	NCT03651271	Biological: Nivolumab Monotherapy Biological: Nivolumab and Ipilimumab and Combination for Metastatic Cancer Biological: Nivolumab and Ipilimumab (3 mg/kg) Combination for Prostate Cancer Biological: Nivolumab and Ipilimumab (5 mg/kg) Combination for Prostate Cancer	Phase 2
Active, but not recruiting	NCT03815942	Biological: ChAdOx1-MVA 5T4 vaccine Drug: Nivolumab [OPDIVO^®^] Infusion	Phase 1 Phase 2
Study discontinued due to drug supply issues; no analysis performed, or meaningful data derived.	NCT01420965	Drug: CT-011 (PD1 Antibody) Other: Sipuleucel-T (PROVENGE^®^) Drug: Cyclophosphamide	Phase 2
Active, but not recruiting	NCT03177460	Biological: Daratumumab (CD38 antibody) Drug: FMS Inhibitor JNJ-40346527 Procedure: Radical Prostatectomy	Phase 1
Recruiting	NCT03805594	Drug: Lu-177-PSMA-617 Biological: Pembrolizumab	Phase 1
Recruiting	NCT04597411	Radiation: 225-Ac-PSMA-617 Radiation: 68-Ga-PSMA-11	Phase 1
Active, but not recruiting	NCT01867333	Biological: PROSTVAC-F/TRICOM Biological: PROSTVAC-V/TRICOM Biological: Enzalutamide (Xtandi)	Phase2
Terminated due to Toxicity	NCT00133224	Biological: Immunotherapy allogeneic GM-CSF secreting cellular vaccine (GVAX) Drug: Chemotherapy (docetaxel and prednisone)	Phase 3
Terminated based on analysis showing <30% chance of patients meeting primary endpoint	NCT00089856	Biological: Imunotherapy with allogeneic prostate vaccine (GVAX) Drug: Chemotherapy (Taxotere and prednisone)	Phase 3
Completed	NCT02234921	Drug: Cyclophosphamide Biological: DRibble Vaccine (targeting DCs) Other name DPV-001 DRibble vaccine Biological: HPV Vaccinations Other Name: Ceravix Drug: Imiquimod	Phase 1
Completed	NCT00140348	Biological: Immunotherapy allogeneic GM-CSF secreting cellular vaccine	Phase 1 Phase 2
Completed	NCT00140400	Biological: Immunotherapy Allogeneic GM-CSF secreting cellular vaccine	Phase 1 Phase 2
Recruiting	NCT03518606	Drug: Durvalumab + Tremelimumab + metronomic Vinorelbine	Phase 1 Phase 2
Recruiting	NCT03493945	Biological: M7824 (bifunctional anti-PD-L1/TGFβ Trap fusion protein) Drug: N-803 Biological: MVA-BN-Brachyury Biological: FPV-Brachyury Drug: Epacadostat	Phase 1 Phase 2
Completed	NCT03384316	Biological: ETBX-051; adenoviral brachyury vaccine Biological: ETBX-061; adenoviral Mucin-1 (MUC1) vaccine Biological: ETBX-011; adenoviral Carcinoembryonic antigen (CEA) vaccine	Phase 1
Active, but not recruiting	NCT02740985	Drug: AZD4635 Drug: Durvalumab Drug: Abiraterone Acetate Drug: Enzalutamide Drug: Oleclumab Drug: Docetaxel	Phase 1
Recruiting	NCT04514484	Drug: Cabozantinib S-malate Biological: Nivolumab	Phase 1
Recruiting	NCT03217747	Biological: OX40 Antibody PF-04518600 Drug: Avelumab Radiation: Radiation Therapy Biological: Utomilumab (4-1BB agonist)	Phase 1 Phase 2
Recruiting	NCT01095848	Biological: DPX-0907 consists of 7 tumor-specific HLA-A2-restricted peptides, a universal T Helper peptide, a polynucleotide adjuvant, a liposome, and Montanide ISA51 VG	Phase 1
Recruiting	NCT03315871	Biological: PROSTVAC-V Biological: PROSTVAC-F Drug: MSB0011359C (M7824) Biological: CV301	Phase 2
Completed	NCT03412786	Bcl-Xl_42-CAF09b vaccine	Phase 1
Completed	NCT02232230	Drug: PROVENGE^®^	Phase 2
Completed	NCT00283829	Drug: docetaxel Drug: IL-2	Phase 1 Phase 2
Completed	NCT02692976	Biological: mDC vaccination Biological: pDC vaccination Biological: mDC and pDC vaccination MUC1, NY-ESO-1, and MAGE-C2	Phase 2
Recruiting	NCT01436968	Biological: Aglatimagene besadenovec + valacyclovir Biological: Placebo + valacyclovir ProstAtack (oncolytic virus)	Phase 3
Active, but not recruiting	NCT03879122	Drug: Ipilimumab 5 MG/ML Drug: Nivolumab 10 MG/ML Drug: Docetaxel Drug: ADT (androgen deprivation therapy)	Phase 2 Phase 3
Active, but not recruiting	NCT01867333	Biological: PROSTVAC-F/TRICOM Biological: PROSTVAC-V/TRICOM Biological: Enzalutamide (Xtandi^®^)	Phase 2

**Table 2 cancers-14-01970-t002:** List of HLA-A*02:01 class-I and HLA-DRB1*01:01 class-II peptides derived from the hPAP42mer sequence.

HHDII/DR1 Mice	Sequence	Haplotype	Length	SYFPEITHI Score hPAP42mer WT	SYFPEITHI Score hPAP42mer Mut
Class I epitopes	SAMTNLAAL	HLA-A*02:01	9 mer	24	30
	ILLWQPIPV		9 mer	24	24
	ALFPPEGVSI	HLA-A*02:01/A*03	10 mer	27/25	27/25
Class II epitopes	LAALFPPEGVSIWNP	HLA-DRB1*01:01	15 mer	25	25
	MSAMTNLAALFPPEG		15 mer	33	33
	PEGVSIWNPILLWQP		15 mer	25	25
	VSIWNPILLWQPIPV		15 mer	25	25
	DRTLMSAMTNLAALF		15 mer	22	30

## Data Availability

The data presented in this study are all available withing the article.

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
