# Peer review of "A Mutated Prostatic Acid Phosphatase (PAP) Peptide-Based Vaccine Induces PAP-Specific CD8^+^ T Cells with Ex Vivo Cytotoxic Capacities in HHDII/DR1 Transgenic Mice"

_cancers, 2022, doi:10.3390/cancers14081970_

Round 1
Reviewer 1 Report
Thank you for submitting your excellent work on an important potential future immunotherapy for prostate cancer patients.
In Table 1, please amend "Lutetium Lu-177-PSMA-617" to "Lu-177-PSMA-617"
("Lutetium" is a duplication of "Lu-177")
If targeted radionuclide therapies such as Lu-177-PSMA are included, consider the inclusion also of Ac-225-PSMA ("ACTION trial")
Author Response
Please, see the attachment.

Reviewer 2 Report
Vaccination is a straightforward way to generate tumor specific T cell responses or boost existing tumor specific T cell responses. Although the cancer vaccination in theory could be very powerful, the expected potential of cancer vaccines has not been realized in the clinical setting. In this manuscript, Pauline Le Vu et al. described a novel cancer vaccine strategy against prostate cancer. By utilizing mutated long PAP peptide and novel adjuvant CAF09, they significantly increased the cancer vaccine efficacy. The vaccine-generated T cells are fully functional and with more memory phenotype subpopulation. This research is interesting and with clinical translational potential.
Listed below are some issues in need of clarification.
Majors:
The efficacy of published PAP-whole protein vaccine should be discussed and compared with the proposed method in discussion section.
Minors:
- In lines 118 and 127, “HLA-A2/HLA-DR1- transgenic mice” and “HHDII/DR1 mice” are mentioned. If they are the same, please specify it.
- In line 136, if the HHDII is just overexpressed, please change the words “knocked in”.
- In line 187, the mutated amino acid should be L, not S.
- Please define the abbreviation “ILL” as “ILLWQPIPV”at its first usage.
- “Samples were acquired on a 10-color Gallios™ flow cytometer (Beckman Coulter) and analyses were performed using Kaluza™ software 1.3 (Beckman Coulter). This sentence has been repeatedly shown at multiple places. Please avoid that.
- In line 369, Fig 6d should be Fig. 1d.
- It is better to present “3.2.1” and “3.2.2” as a figure or supplementary figure.
- Figure 3A, the CAF09 adjuvant group showed less IFNgamma production in responding to some peptide stimulation. This should be mentioned and discussed.
- Figure 4d, “TNFa” and “IFNg” need to use Roman alphabet.
- In lines 529 and 530 “Vaccination induces secretory and cytotoxic ILL-specific cytotoxic T lymphocytes (CTLs) displaying an exhausted phenotype”. Since these CTLs are fully functional as shown in Figure 5, the expression of PD-1, LAG-3 shouldn’t been consider as exhaustion markers. When T cells get activated, these makers are expressed as well. It is better to change the sentence to “Vaccination induces secretory and cytotoxic ILL-specific cytotoxic T lymphocytes (CTLs) ex vivo” or similar.
Author Response
Please, see the attachment

Reviewer 3 Report
Changes in work are needed to improve the quality of work
1. the introduction is too extensive it should be shortened and focused on the essentials
2. in the section materials and methods for all reagents and equipment it is necessary to write the country of manufacture
3. for the method of determination of cytokines from produced cells at the end of paragraph (line 211) add a reference as suggested in previous works on the possibilities of determination of cytokines from produced cells:
Multiomic analysis of cytokines in immuno-oncology. Expert Rev Proteomics. 2020 Sep; 17 (9): 663-674.
4. For flow cytometry and determination of apoptosis, add the method at the end of the paragraph: TNF-alpha induced apoptosis is accompanied by rapid CD30 and slower CD45 shedding from K-562 cells. J Membr Biol. 2011 Feb; 239 (3): 115-22.
5. for the method of determining cytotoxicity with the help of radioactive chromium, add a reference that has shown that the method is still valid today and cite the publication: Clinical stage-depending decrease in NK cell activity in multiple myeloma patients. With Oncol. 2007; 24 (3): 312-7.
6. In the discussion for showing immunosuppression in cancer, we need to use references other than reference 29 because that reference refers to Autoimmunity. Add references showing that the value of cytokines and other proinflammatory mediators is increased in the tumor environment. these mediators induce suppression of effector CD8 and NK cells, otherwise necessary in tumor elimination as shown previously: The role of cytokines in the regulation of NK cells in the tumor environment. Cytokine. 2019 May; 117: 30-40.
7. In the discussion, explain a little better the reason why there was no significant increase in CD62L- memory cells.
8. There are various markers to confirm long-lived memory cells, such as CD127 in combination with IL-2 receptor and discuss marker selection options
Author Response
Please, see the attachment.

Round 2
Reviewer 3 Report
partiaally replay